# Soloxolone *N*-3-(Dimethylamino)propylamide Restores Drug Sensitivity of Tumor Cells with Multidrug-Resistant Phenotype via Inhibition of P-Glycoprotein Efflux Function

**DOI:** 10.3390/molecules29204939

**Published:** 2024-10-18

**Authors:** Arseny D. Moralev, Oksana V. Salomatina, Nariman F. Salakhutdinov, Marina A. Zenkova, Andrey V. Markov

**Affiliations:** 1Institute of Chemical Biology and Fundamental Medicine, Siberian Branch of the Russian Academy of Sciences, 630090 Novosibirsk, Russia; arseniimoralev@gmail.com (A.D.M.); ana@nioch.nsc.ru (O.V.S.); marzen@niboch.nsc.ru (M.A.Z.); 2N.N. Vorozhtsov Novosibirsk Institute of Organic Chemistry, Siberian Branch of the Russian Academy of Sciences, 630090 Novosibirsk, Russia; anvar@nioch.nsc.ru

**Keywords:** ABC transporters, cancer, chemotherapy, multidrug resistance, pentacyclic triterpenoids, P-glycoprotein, soloxolone

## Abstract

Multidrug resistance (MDR) remains a significant challenge in cancer therapy, primarily due to the overexpression of transmembrane drug transporters, with P-glycoprotein (P-gp) being a central focus. Consequently, the development of P-gp inhibitors has emerged as a promising strategy to combat MDR. Given the P-gp targeting potential of soloxolone amides previously predicted by us by an absorption, distribution, metabolism, excretion, and toxicity (ADMET) analysis, the aim of the current study was to experimentally verify their P-gp inhibitory and MDR reversing activities in vitro. Screening of soloxolone amides as modulators of P-gp using molecular docking and cellular P-gp substrate efflux assays revealed the ability of compound **4** bearing a *N*-3-(dimethylamino)propylamide group to interact with the active site of P-gp and inhibit its transport function. Blind and site-specific molecular docking accompanied by a kinetic assay showed that **4** directly binds to the P-gp transmembrane domain with a binding energy similar to that of zosuquidar, a third-generation P-gp inhibitor (ΔG = −10.3 kcal/mol). In vitro assays confirmed that compound **4** enhanced the uptake of Rhodamine 123 (Rho123) and doxorubicin (DOX) by the P-gp-overexpressing human cervical carcinoma KB-8-5 (by 10.2- and 1.5-fold, respectively (*p* < 0.05, unpaired *t*-test)) and murine lymphosarcoma RLS40 (by 15.6- and 1.75-fold, respectively (*p* < 0.05, unpaired *t*-test)) cells at non-toxic concentrations. In these cell models, **4** showed comparable or slightly higher activity than the reference inhibitor verapamil (VPM), with the most pronounced effect of the hit compound in Rho123-loaded RLS40 cells, where **4** was 2-fold more effective than VPM. Moreover, **4** synergistically restored the sensitivity of KB-8-5 cells to the cytotoxic effect of DOX, demonstrating MDR reversal activity. Based on the data obtained, **4** can be considered as a drug candidate to combat the P-gp-mediated MDR of tumor cells and semisynthetic triterpenoids, with amide moieties in general representing a promising scaffold for the development of novel therapeutics for tumors with low susceptibility to antineoplastic agents.

## 1. Introduction

Despite significant advances in cancer therapy, chemotherapy remains the “gold” standard and commonly used strategy to improve overall patient survival [1,2,3]. In many cases, the high efficacy of chemotherapeutic drugs in the early stages of therapy is replaced by a significant decrease in the sensitivity of tumor cells to the treatment, not only to initial first-line drugs but also to other structurally different therapeutics (the phenomenon known as a multidrug resistance (MDR)) [4,5]. One of the key mechanisms of MDR induction in tumor cells is related to the activity of transmembrane transporters of the ATP-binding cassette (ABC) family, resulting in the depletion of the intracellular concentration of chemotherapeutic drugs due to their active removal from the cells [6,7,8]. The most studied ABC transporter is P-glycoprotein (P-gp), which exhibits broad substrate specificity for xenobiotic molecules, including chemotherapeutic agents used in cancer treatment [9,10]. Given the pivotal role of P-gp in MDR induction, the development of P-gp inhibitors is emerging as a promising strategy to resensitize tumor cells to chemotherapy [11,12].

The P-gp inhibitors developed to date can be divided into three generations [13], among which the first- and second-generation anti-P-gp compounds exhibit pronounced toxicity, poor affinity, and unpredictable drug–drug interactions [14]. Third-generation P-gp inhibitors, despite their high specificity for P-gp and marked suppressive effect on P-gp-mediated drug efflux in vitro and in vivo, also have drawbacks, showing unsatisfactory results in clinical trials due to adverse toxicity and lack of MDR reversing activity [14]. Considering these shortcomings, there are currently no P-gp inhibitors approved for clinical use on the pharmacological market, suggesting the advisability of searching for new drug development strategies in this field.

Currently, natural compounds and their semisynthetic derivatives are considered the fourth generation of P-gp inhibitors due to their high anti-MDR potential and lower systemic toxicity compared to chemically synthesized drugs [15,16]. Pentacyclic triterpenoids represent a large class of plant metabolites with diverse antitumor activities [17,18,19], including synergism in combination with known antitumor drugs [20,21,22]. Despite the latter fact, to the best of our knowledge, studies demonstrating the P-gp inhibitory activity of pentacyclic triterpenoids are currently limited [23,24,25,26,27,28] and require further close attention.

Previously, our research group reported the synthesis and evaluation of the antitumor potency of C-30 amide-containing derivatives of soloxolone methyl, a cyanoenone-bearing semisynthetic triterpenoid [20,29]. We have shown that representatives of this series of derivatives not only have the ability to increase the sensitivity of glioblastoma cells to temozolomide [20], but also, based on chemoinformatics data, they can be hypothetically considered as potential inhibitors of P-gp [29]. The present work aims to test this hypothesis and to answer the question whether cyanoenone-containing triterpenoids can directly block the pump function of P-gp. To address this issue, the interaction of soloxolone amides bearing various aliphatic (**1**–**4**) and aromatic (**5**–**8**) substituents (Figure 1) with the transmembrane domain of P-gp was evaluated by a molecular modeling approach, followed by a detailed verification of the obtained in silico data in MDR-related cell models. The obtained data significantly extend our understanding of the anti-P-gp potential of both pentacyclic triterpenoids in general and cyanoenone-bearing triterpenoids in particular, and the identified lead compound **4**—soloxolone *N*-3-(dimethylamino)propylamide [*N*-(3′-(Dimethylamino)propyl)-2-cyano-3,12-dioxo-18βH-olean-9(11),1(2)-dien-30-amide] can be considered as a drug candidate for controlling the MDR status of tumor cells in cancer.

## 2. Results and Discussion

### 2.1. Soloxolone Amides 1–8 Can Interact with the Transmembrane Domain of P-gp

First, to independently assess the P-gp targeting potential of soloxolone amides **1**–**8**, their anti-P-gp properties were predicted using three independent absorption, distribution, metabolism, excretion, and toxicity (ADMET) web servers, AdmetSAR2.0 [30], PgpRules [31], and vNN-ADMET [32]. As shown in Figure 2A, all prediction tools used showed a high potential of **1**–**8** to suppress P-gp activity.

Second, to verify these data, the ability of **1**–**8** to bind to the active site of P-gp was accessed by molecular docking. The results obtained indicated that all the investigated molecules could directly interact with the transmembrane domain of P-gp, with low Gibbs free energies (mean ΔG = −10.6 kcal/mol) comparable to that of zosuquidar (ZSQ), a known P-gp inhibitor (ΔG^ZSQ^ = −10.4 kcal/mol) (Figure 2B,C). Considering that binding energies lower than −7 kcal/mol indicate the formation of relatively stable protein–ligand complexes [33], the calculated ΔG values of the tested triterpenoids showed the relevance of their direct P-gp targeting potential.

It was found that the aryl-bearing derivatives **5**–**7** demonstrated the lowest binding energies among the analyzed molecules, probably due to the presence of π–π stacking between their aromatic moieties and Phe728 (**5**,**6**) or Phe303 (**7**) in addition to conventional hydrogen bonds (H-bonds) (Figure 2D). Interestingly, the aromatic moiety of **8**, which also showed high affinity to P-gp, does not play a critical role in the positioning of the molecule in the transmembrane domain of P-gp, probably due to the longer hydrocarbon linker. The observed low binding energy of **8** can be explained by the presence of two opposite H-bonds markedly stabilizing the protein–ligand complex (Figure 2D).

To note, all obtained docking complexes contained H-bonds, but the side groups in most of the compounds studied did not participate in their formation. Compound **3** was found to be the most enriched in H-bonds, with three H-bonds between carbonyl groups at positions C-3, C-12, and C-30 and Tyr307, Gln990, and Trp232, respectively (Figure 2D). Compounds **1** and **8** formed two opposite H-bonds at positions C-3 and C-30, securely anchoring these molecules in the transmembrane domain of P-gp. The docking complex of compound **7** also contained two H-bonds, but their location was limited only by the side moiety of the triterpenoid. The other compounds formed only one H-bond each, which, despite the observed low binding energies (Figure 2C), may indicate the low stability of the obtained docking complexes.

Based on the results of in silico studies, the lead compounds **2** and **4**–**8** with a ΔG of less than −10 kcal/mol were selected for further experimental verification.

### 2.2. Compound 4 Enhances the Intracellular Accumulation of Rhodamine 123 in KB-8-5 Cells That Overexpress P-gp

To verify the P-gp inhibitory activity of compounds **2** and **4**–**8**, human cervical carcinoma KB-8-5 cells overexpressing P-gp [22] were used. First, in order to select low-toxicity compounds and identify their working concentration range, the toxicity of **2** and **4**–**8** against KB-8-5 cells after 30 min incubation (time period used for further evaluation of P-gp-mediated Rhodamine 123 uptake by MDR cells [34,35]) was determined by an MTT assay. As shown in Figure 3A,B, three out of six lead compounds (**5**, **6**, **8**) did not affect cell viability up to 100 µM, and **4** was non-toxic for KB-8-5 cells up to 40 µM. The remaining triterpenoids (**2** and **7**) had a significant detrimental effect on cell viability starting at 20 µM (Figure 3A,B) and were excluded from further analysis. Considering that concentrations in the range of 20–50 µM are acceptable for the study of P-gp inhibitory activity of natural-based compounds [24,36], the anti-P-gp effect of **4**–**6** and **8** at 40 µM was further evaluated.

Surprisingly, the 30 min incubation of KB-8-5 cells with Rhodamine 123 (Rho123), a known fluorescent P-gp substrate, in combination with the tested compounds, revealed that only **4** significantly induced the intracellular accumulation of Rho123 (Figure 3C,D). It was found that **4** increased the number of Rho123^high^ cells and the mean intracellular Rho123 fluorescence by 18.4- and 10.2-fold, respectively, compared to the untreated control (Figure 3C,D), and this effect of **4** was dose-dependent (Figure 3E) and comparable to verapamil (VPM), a known P-gp inhibitor (Figure 3C–E). Unexpectedly, despite the low binding energies with the transmembrane domain of P-gp (Figure 2B), **5**, **6**, and **8** had no significant effect on Rho123 accumulation in KB-8-5 cells (Figure 3C,D). We hypothesize that the observed differences in the efficiency of inhibition of Rho123 efflux by the studied compounds may be due to the peculiarities of their direct interaction with P-gp. In contrast to the inactive compounds **5**, **6**, and **8**, compound **4**, which shows pronounced anti-P-gp activity, forms a more stable complex with P-gp containing three H-bonds (Figure 2C), including interactions with Gln990 and Trp232, which play a crucial role in the positioning of various P-gp inhibitors [37,38,39,40]. Moreover, a comparison of the molecular surfaces of the tested compounds and ZSQ within docking complexes revealed that **4** occupies a more similar position to ZSQ in the transmembrane domain of P-gp compared to other compounds (Figure 3F), which may also explain the high anti-P-gp potential of **4**. Further molecular dynamics studies are required to test these hypotheses in more detail.

Note that the low level of toxicity of **4** observed in the 30 min experiment described above (Figure 3A) was maintained when the incubation time of KB-8-5 cells with **4** was increased up to 72 h (Figure 3G), indicating the feasibility of further investigation of **4** as a P-gp inhibitor. Moreover, the additional evaluation of Rho123 accumulation in a series of non-MDR tumor cell lines, namely KB-3-1 [23], A549 [41], and B16 [42], revealed no significant enhancement in the intracellular Rho123 signal in compound-**4**-treated cells (Figure 3H,I), confirming the P-gp-targeting effect of **4**.

### 2.3. Compound 4 Increases Cytotoxicity and Intracellular Accumulation of Doxorubicin in KB-8-5 Cells

Since high P-gp efflux activity is a major cause of MDR of tumor cells [9,10], the ability of **4** to reverse the resistance of KB-8-5 cells to doxorubicin (DOX), an anthracycline drug and known P-gp substrate, was further investigated. Similar to the Rho123 assay data, **4** effectively suppressed DOX efflux from KB-8-5 cells, increasing the number of DOX^high^ cells and the average intracellular DOX fluorescence by 9.1-fold (Figure 4A) and 1.5-fold (Figure 4B), respectively, compared to the control. The efficacy of **4** was comparable to VPM.

Consistent with these data, **4** significantly increased the sensitivity of model cells to DOX cytotoxicity. It was shown that co-incubation of KB-8-5 cells with **4** and DOX for 72 h (time period at which significant DOX cytotoxicity is achieved [23]) has a pronounced synergistic effect (Figure 4C,D), with the highest Bliss synergy score of 21.8 when **4** and DOX were combined at 20 μM and 2 μM, respectively (Figure 4D).

Considering the reported ability of natural-based compounds to reverse the MDR phenotype of tumor cells by suppressing P-gp expression rather than through direct interaction with P-gp [43,44,45], the effect of **4** on P-gp expression in KB-8-5 cells was further evaluated. The performed RT-PCR, Western blot, and flow cytometry analysis clearly demonstrated the absence of a statistically significant effect of **4** on the mRNA and protein levels of P-gp in KB-8-5 cells at both 2 and 72 h (Figure 4E–G). These results, together with the results of molecular modeling (Figure 2C) and **4**-mediated blockade of Rho123 (Figure 3C) and DOX (Figure 4A) efflux, confirmed the direct interaction of **4** with P-gp.

### 2.4. Compound 4 Binds to the Modulation Site of the P-gp Transmembrane Domain

As shown above, the inhibitory effect of **4** on Rho123 efflux (Figure 3C,D) was more pronounced compared to DOX efflux (Figure 4A,B), which may be related to the site to which **4** binds in P-gp. Based on published reports, the transmembrane domain of P-gp contains three possible binding sites, namely the modulation site (M), which is specific for known effective P-gp inhibitors (zosuquidar, tariquidar, elacridar, etc.), and two polyspecific recognition sites R and H, which bind P-gp substrates (Figure 5A) [33,46]. To elucidate the specific features of the interaction of **4** with P-gp, site-specific molecular docking was performed. The transmembrane domain of P-gp was divided into three grid boxes covering the M-, R-, and H-sites (Figure 5A), and ligands (Rho123, DOX, ZSQ, and **4**) were separately docked into them. The obtained results clearly showed that DOX preferably interacted with the R-site (ΔG_DOX_^R^ = −9.3 kcal/mol), whereas Rho123 was less selective and bound to the R- and M-sites with similar potency (ΔG_Rho123_^R^ = −8.4 kcal/mol, ΔG_Rho123_^M^ = −8.3 kcal/mol) (Figure 5A). The H-site was found to be the least favorable compartment for interaction with the indicated P-gp substrates, which is in agreement with previously published data [33,47]. In the case of P-gp inhibitors, ZSQ and **4** appeared to be able to interact with all the binding sites analyzed, while showing some specificity for the M-site, to which the compounds bound with the lowest free Gibbs energies (ΔG_ZSQ_^M^ = −10.7 kcal/mol, ΔG**_4_**^M^ = −10.4 kcal/mol) (Figure 5B).

To verify these data, an independent blind docking covering the entire transmembrane domain of P-gp was performed. A comparison of the ligand conformations obtained by blind docking (Figure 5A, yellow-colored molecules) and site-specific docking (Figure 5A, red- or blue-colored molecules) revealed their high overlap and close free Gibbs energies, confirming the preferential binding of DOX to the R-site, the non-selective binding of Rho123 to the R- and M-sites, and the targeting of ZSQ and **4** to the M-site discussed above.

To confirm the results obtained in silico, a kinetic study of the intracellular accumulation of Rho123 and DOX in compound-**4**-treated KB-8-5 cells was performed. A Lineweaver–Burk analysis revealed that the P-gp inhibitory effect of **4** included a dose-dependent competitive inhibition of Rho123 transport (V_max_ was unaffected, whereas K_M_ was decreased after incubation with **4** at 10 μM and 20 μM by 1.3- and 2.7-fold, respectively) (Figure 5C) and a non-competitive inhibition of DOX efflux (V_max_ and K_M_ were both increased after incubation with **4** at 10 μM (by 1.5- and 1.6-fold, respectively) and 20 μM (by 4.0- and 4.8-fold, respectively)) (Figure 5D). Thus, the kinetic data confirm the preferential binding of **4** to the M-site of P-gp, as predicted above by molecular modeling.

### 2.5. Verification of P-gp Inhibitory Activity of 4 in RLS40 Cells

Finally, to double check the inhibitory effect of **4** on P-gp transport activity, compound-**4**-induced changes in Rho123 and DOX efflux were independently evaluated in murine lymphosarcoma RLS40 cells overexpressing P-gp [48]. First, a non-toxic concentration of **4** at 20 μM was detected in RLS40 cells by a WST assay (Figure 6A) and used for further experiments. The Rho123 assay showed a significant increase in Rho123^high^ cell population and intracellular Rho123 accumulation in compound-**4**-treated RLS40 cells by 21.3-fold and 15.6-fold, respectively, compared to the untreated control (Figure 6B,C). Consistent with these data, **4** effectively inhibited DOX efflux from RLS40 cells, as evidenced by an increase in the number of DOX^high^ cells and intracellular DOX accumulation of 93.2% and 75%, respectively, compared to the control (Figure 6D,E). Notably, the efficacy of **4** was comparable to or slightly higher than that of VPM in the DOX (Figure 6E) and Rho123 (Figure 6C) assays performed. Intriguingly, the anti-P-gp activity of **4** in RLS40 cells (Figure 6) was more pronounced compared to that in KB-8-5 cells (Figure 3C–E and Figure 4A,B), which may be explained by the different levels of P-gp expression in these cells and possibly by the different affinity of **4** for human and murine P-gp, which requires further detailed studies. In any case, the demonstrated inhibitory activity of **4** against P-gp-mediated efflux in two independent cell models suggests its high therapeutic potential in MDR tumors.

### 2.6. Limitations of the Study

Despite the data obtained clearly demonstrating the potent anti-P-gp activity of **4**, this study has a number of limitations. First, given the difficulty in correctly interpreting the stability of ligand–P-gp complexes obtained by molecular docking, additional molecular dynamic analysis is desirable. Second, it would be advisable to compare the efficacy of the P-gp inhibitory activity of **4** not only with VPM but also with representatives of the third generation of P-gp inhibitors (ZSQ or its analogues), since **4** and VPM interact with different binding sites, namely the M-site (see Section 2.4) and the nucleotide-binding domain of P-gp [49,50], respectively. Third, given the unpredictable side effects of known P-gp inhibitors [14] and the overexpression of P-gp in a number of normal tissues, including the liver, kidney, and blood–brain barrier [51], evaluation of the cytotoxicity of **4** against non-transformed cells as well as its safety in animal models is required. Fourth, considering the impact of not only P-gp but also other ABC family transporters, such as MRP1 and BCRP, in MDR development [52], the modulating effect of **4** on their pump activity should be further evaluated. Finally, the pronounced anti-P-gp activity of **4** demonstrated here should be further tested on MDR cells of other origins for a more comprehensive analysis of its MDR reversal potency.

## 3. Materials and Methods

The synthesis of soloxolone amides (**1**–**8**) and their physicochemical characteristics are presented in the work [29].

### 3.1. In Silico Prediction of P-glycoprotein Inhibitor Specificity of Soloxolone Amides

The ability of **1**–**8**, DOX, VPM, and ZSQ to act as inhibitors of P-glycoprotein was predicted by a panel of web tools, including AdmetSAR 2.0 (http://lmmd.ecust.edu.cn/admetsar2/, (accessed on 2 September 2024)), PgpRules (https://pgprules.cmdm.tw/, (accessed on 2 September 2024)), and vNN-ADMET (https://vnnadmet.bhsai.org/vnnadmet/, (accessed on 2 September 2024)), based on the chemical structures of the molecules analyzed.

### 3.2. Cell Lines and Evaluated Compounds

Human lung adenocarcinoma A549 cells, human cervical carcinoma KB-3-1 cells, and murine melanoma B16 cells were obtained from the Russian Cell Culture Collection (St. Petersburg, Russia). Human cervical carcinoma KB-8-5 cells with an MDR phenotype were generously donated by Prof. M. Gottesman (National Institutes of Health, Bethesda, USA). KB-8-5, KB-3-1, A549, and B16 cells were grown in Dulbecco’s modified Eagle’s medium (DMEM) (Sigma-Aldrich, St. Louis, MO, USA). Murine lymphosarcoma RLS40 cells with an MDR phenotype were purchased from the cell bank of the Institute of Chemical Biology and Fundamental Medicine SB RAS (Novosibirsk, Russia) and were grown in Iscove’s Modified Dulbecco’s Medium (IMDM) (Sigma-Aldrich, St. Louis, MO, USA). All culture media contained 10% (*v*/*v*) heat-inactivated fetal bovine serum (FBS) (Dia-M, Moscow, Russia), penicillin at 10,000 IU/mL, streptomycin at 10,000 μg/mL, and amphotericin at 25 μg/mL (MP Biomedicals, Illkirch-Graffenstaden, France). In addition, cells were cultured in the presence of 300 and 40 nM vinblastine for KB-8-5 and RLS40 cells, respectively. The cells were incubated at 37 °C in 5% CO_2_ (hereafter, standard conditions). Soloxolone amides were dissolved in DMSO (stock solution: 10 mM) and stored at −20 °C until the experiments.

### 3.3. Evaluation of Cytotoxicity by MTT Assay

KB-8-5 cells were seeded in 96-well plates at a density of 1 × 10^4^ cells/well (n = 4), incubated for 24 h under standard conditions, and then treated with compounds (10–100 µM) for 30 min or 72 h. Thereafter, 10 µL of MTT at 5 mg/mL (Sigma-Aldrich, St. Louis, MO, USA) was added to each well, and formazan crystals formed after a 2 h incubation under standard conditions were dissolved with DMSO. Finally, the level of cell viability in the compound-treated groups was evaluated compared to the control (the cells incubated with the corresponding volume of DMSO) by measuring the absorbance at 570 and 620 nm using a Multiscan RC plate reader (Thermo LabSystems, Helsinki, Finland).

### 3.4. Evaluation of Cytotoxicity by Water-Soluble Tetrazolium (WST) Test

Since RLS40 cells are characterized by low adherence to culture plastic [53], the water-soluble tetrazolium (WST) test was used to assess compound toxicity. RLS40 cells were seeded in quadruplicate in 96-well plates at a density of 4 × 10^4^ cells/well. The cells were incubated for 24 h under standard conditions, followed by treatment with the triterpenoids (10–80 µM) for 72 h. Thereafter, 10 μL of WST-1 at 0.5 mg/mL (Roche, Basel, Switzerland) was added to each well, and the cells were incubated for an additional 3 h with WST-1 under standard conditions. Absorbance was determined at 450 and 620 nm using a Multiscan RC plate reader, followed by calculation of the cell viability rate in the experimental groups compared to the control.

### 3.5. Evaluation of the Combined Cytotoxicity of 4 and Doxorubicin

To explore the effect of **4** on doxorubicin (DOX) cytotoxicity, KB-8-5 cells were seeded into 96-well plates at 1 × 10^4^ cells per well and incubated overnight under standard conditions in DMEM with 10% fetal bovine serum. Subsequently, the medium was replaced with DMEM containing **4** (2.5–20 µM) and/or DOX (0.5–32 µM) for 72 h. The number of viable cells was determined by the MTT assay as described above, and the combined effect of **4** and DOX was further analyzed using SynergyFinder v.3.0 software (https://synergyfinder.fimm.fi/; Bliss model, (accessed on 3 June 2024)).

### 3.6. Rhodamine 123 and Doxorubicin Accumulation Assay

Cells were seeded in 24-well plates at 10^5^ cells/well (n = 3) for 24 h, followed by incubation with 5.25 μM Rhodamine 123 (Rho123) or 2 µM DOX for 30 min or 2 h, respectively, with or without the tested compounds (40 μM for KB-8-5, KB-3-1, A549, and B16 cells; 20 μM for RLS40 cells) or verapamil (VPM) at 50 μM. After the incubation period, cells were washed twice with PBS, harvested with a TrypLE Express (Gibco, Grand Island, NY, USA), and resuspended in fresh medium. The intracellular accumulation of Rho123 and DOX was assessed by the level of their autofluorescence using a NovoCyte Flow Cytometer (ACEA Biosciences, San Diego, CA, USA) with 10,000 events for each sample.

### 3.7. Kinetic Characterization for P-gp Inhibition

The 5 × 10^5^ KB-8-5 cells, plated in triplicate in sterile 1.5 mL tubes, were incubated in medium containing Rho123 or DOX (2.5–40 µM) with or without (control) **4** (10 or 20 µM) under standard conditions for 150 min (in the case of DOX) or 30 min (in the case of Rho123). After incubation, the cells were washed twice with ice-cold PBS and lysed with lysis buffer (0.75 M HCl and 0.2% Triton-X100 in isopropanol) for 15 min under standard conditions. The lysate from each tube was then transferred to black-walled 96-well plates in triplicate. The fluorescence level of DOX and Rho123 in the lysate was determined using a CLARIOstar Plus microplate reader (BMG Labtech, Offenburg, Germany). The relationship between the intracellular accumulation of DOX or Rho123 and the concentration of **4** over time was analyzed by Lineweaver–Burk plots.

### 3.8. Assessment of ABCB1 Expression by Real-Time Quantitative Polymerase Chain Reaction (RT-qPCR)

Total RNA was extracted from KB-8-5 cells after treatment with compound **4** (40 µM) using TRIzol reagent (Ambion, Austin, TX, USA). Reverse transcription of 4 μg of total RNA was performed using 5× RT buffer, M-MuLV-RH-revertase (Biolabmix, Novosibirsk, Russia), and the oligo (dT) primers according to the manufacturer’s protocol. qRT-PCR was performed using HS-qPCR (2×) master mix (Biolabmix, Novosibirsk, Russia), DNA matrix, and forward and reverse primers (Table 1) as it was described in our previous work [23]. *GAPDH* was used as an internal control; relative expression was calculated by the ΔΔCt method.

### 3.9. Western Blotting

For the Western blot analysis, KB-8-5 cells were lysed in the Laemmli buffer (Sigma-Aldrich, St. Louis, MO, USA). The protein samples were separated on an SDS-PAGE gel and transferred to a polyvinylidene fluoride (PVDF) membrane (Millipore, Burlington, MA, USA). The membranes were blocked by incubation with 2% nonfat dry milk for 1 h and then incubated with primary monoclonal antibodies against P-gp (P7965, 1:750, Sigma-Aldrich, St. Louis, MO, USA) and GAPDH (A19056, 1:5000, ABclonal, Wuhan, China). The membranes were then washed and incubated with secondary horseradish peroxidase-conjugated goat anti-mouse (A9917, 1:5000, Sigma-Aldrich, St. Louis, MO, USA) or goat anti-rabbit (ab6721, dilution: 1:3000, Abcam, Waltham, MA, USA) IgG for 1 h, washed again, and incubated with a Western Blotting Chemiluminescent Reagent Kit (Abcam, Cambridge, MA, USA) for 5 min. The intensity of the bands was quantified using iBright Analysis Software v. 5.1.0 (Thermo Fisher Scientific, Waltham, MA, USA).

### 3.10. Evaluation of P-gp Expression by Flow Cytometry

The KB-8-5 cells were seeded at a density of 5 × 10^4^ cells per well in 24-well plates and incubated under standard conditions for 24 h, after which the medium was replaced with a fresh medium containing **4** (40 μM). After 2 or 72 h of incubation, the cells were washed twice with PBS, digested with TrypLE Express, resuspended in 1.5 mL tubes, and centrifuged at 400× *g* for 5 min. The supernatant was removed, and the cells were incubated with mouse monoclonal anti-P-gp antibody (P7965, 1:300, Sigma-Aldrich, St. Louis, MO, USA) for 30 min. The cells were then washed with PBS and incubated with fluorescent anti-mouse IgG (A21202, 1:200, Thermo Fisher Scientific, Waltham, MA, USA). Finally, the cells were washed, and their fluorescence was analyzed using a NovoCyte Flow Cytometer, recording 10,000 events for each sample.

### 3.11. Molecular Docking

Molecular docking of ligand binding to the transmembrane domain of P-gp was performed using AutoDock Vina [54]. The crystal structure of ABCB1 with the inhibitor zosuquidar (ZSQ) (PDB ID: 7A6F) was downloaded from the RCSB Protein Data Bank (https://www.rcsb.org/, (accessed on 3 June 2024)). Water molecules and cocrystallized ZSQ were removed, and polarized hydrogens and Gasteiger charges were added using AutoDockTools v.1.5.7. Ligand structures (**4**, DOX, Rho 123, and ZSQ) were generated using ChemDraw 12.0 and converted to 3D conformation using Marvin Sketch v.5.12 followed by the optimization of their geometry using Avogadro v.1.2.0 (MMFF94 force field). The grid boxes were set as follows: for the full transmembrane domain, 60 × 60 × 60 Å centered at [x, y, z = 164.475, 155.155, 161.687]; for the H-site, 40 × 40 × 40 Å centered at [x, y, z = 166.151, 148.487, 170.368]; for the R-site, 40 × 40 × 40 Å centered at [x, y, z = 161.045, 168.066, 164.535]; and for the M-site, 40 × 40 × 40 Å centered at [x, y, z = 164.665, 150.693, 152.239]. Other parameters were set to their default values. Discovery Studio Visualizer 17.2.0 (Dassault Systèmes, Cedex, France) was used to analyze and visualize the docking results and to calculate the volume overlap between the top-ranked conformations of the investigated compounds and the reference P-gp inhibitor ZSQ.

### 3.12. Statistical Analysis

A statistical analysis was performed using Microsoft Excel (Microsoft, Redmond, WA, USA), with a significance level of changes at *p* < 0.05. The normality of the distribution of the analyzed samples was assessed by the Shapiro–Wilk test. The unpaired Student’s *t*-test was used to assess statistical significance between the experimental groups and the control. All experiments were performed in 3–4 biological replicates.

## 4. Conclusions

Taken together, our results clearly demonstrate that soloxolone *N*-3-(dimethylamino)propylamide **4** [*N*-(3′-(Dimethylamino)propyl)-2-cyano-3,12-dioxo-18βH-olean-9(11),1(2)-dien-30-amide] exhibits pronounced anti-P-gp activity in vitro. It was found that **4**, without affecting P-gp expression, effectively suppressed P-gp transport function in human KB-8-5 and mouse RLS40 cells with an MDR phenotype, as manifested by a significant increase in cellular uptake of known P-gp substrates, namely Rho123 and DOX. This effect mediated a pronounced synergistic cytotoxic effect of **4** in combination with DOX in KB-8-5 cells. Molecular docking studies accompanied by kinetic analysis revealed that **4** preferentially interacts with the M-site of the P-gp transmembrane domain. Given the comparable potency of **4** with VPM, a known P-gp inhibitor, compound **4** can be considered as a novel promising drug candidate for the sensitization of tumors with an MDR phenotype to chemotherapy.

## Figures and Tables

**Figure 1 molecules-29-04939-f001:**
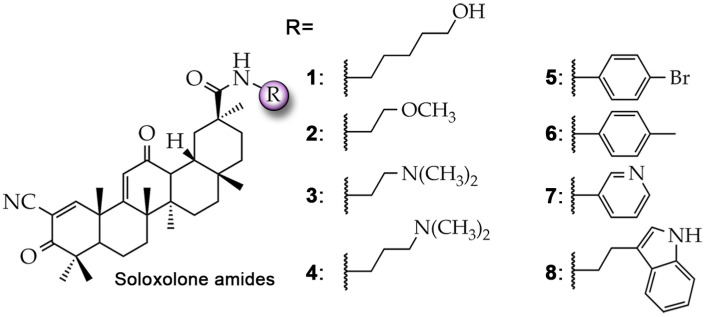
Structures of soloxolone amides **1**–**8** investigated in this study.

**Figure 2 molecules-29-04939-f002:**
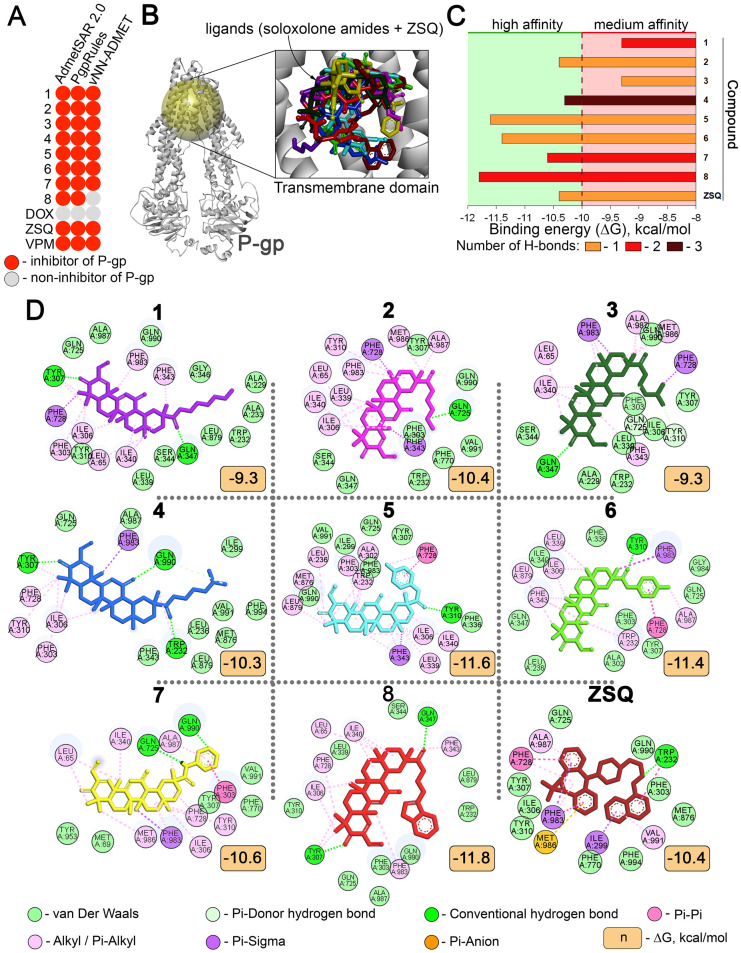
In silico analysis of P-gp transmembrane domain targeting with soloxolone amides. (**A**) In silico prediction of P-gp inhibitory potential of soloxolone amides **1**–**8**. (**B**) Three-dimensional representation of binding poses of soloxolone amides **1**–**8** and ZSQ in transmembrane domain of P-gp (shown as yellow sphere). (**C**) Binding energies of **1**–**8** and ZSQ with transmembrane domain of P-gp and number of H-bonds in the resulting docking complexes. (**D**) Two-dimensional representation of predicted binding poses between **1**–**8**, ZSQ, and transmembrane domain of P-gp visualized using Discovery Studio Visualizer.

**Figure 3 molecules-29-04939-f003:**
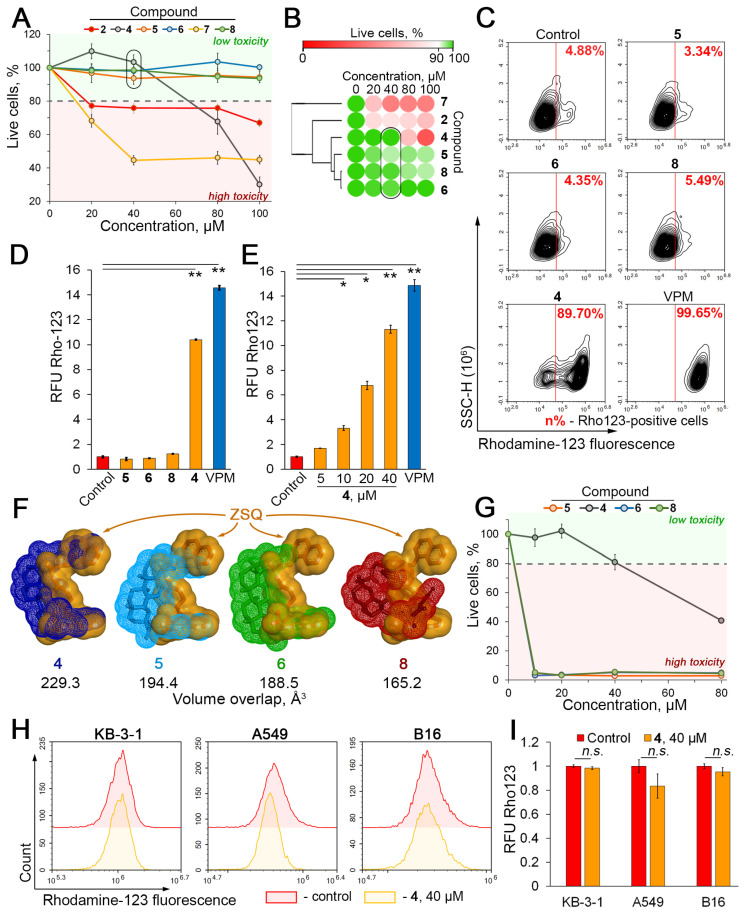
Effect of soloxolone amides on Rhodamine-123 (Rho123) accumulation by KB-8-5 cells. (**A**) Cytotoxicity of **2** and **4**–**8** (20–100 μM) in KB-8-5 cells evaluated by MTT assay after 30 min of incubation. Non-toxic compounds and their concentration are highlighted with a rounded rectangle. (**B**) Heat map showing sensitivity of KB-8-5 cells to **2** and **4**–**8** (20–100 μM) evaluated by MTT test after 30 min of incubation. Non-toxic compounds and their concentrations are highlighted with a rounded rectangle. (**C**) Flow cytometric analysis of Rho123 fluorescence of KB-8-5 cells after incubation with Rho123, VPM (50 μM), **4**–**6**, and **8** (40 μM) for 30 min. Control, untreated cells stained with Rho123 only. (**D**) Relative fluorescence level of KB-8-5 cells incubated with Rho123, VPM (50 μM), **4**–**6**, and **8** (40 μM) for 30 min (n = 3). Control, untreated cells stained with Rho123. (**E**) Dose-dependent effect of **4** (5–40 μM) on Rho123 uptake by KB-8-5 cells (n = 3). (**F**) Three-dimensional representation of **4**–**6** and **8** surfaces overlapping with ZSQ (yellow) in transmembrane domain of P-gp. Conformations were predicted using AutoDock Vina. Visualization and calculation of volume overlap were performed using BIOVIA Discovery Studio. (**G**) Cytotoxicity of **4**–**6** and **8** (10–80 μM) in KB-8-5 cells evaluated by MTT assay after 72 h incubation. (**H**) Flow cytometry assessment of Rho123 uptake by KB-3-1, A549, and B16 cells after incubation with Rho123 and **4** (40 μM) for 30 min. Control, untreated cells stained with Rho123 only. (**I**) Relative fluorescence level of KB-3-1, A549, and B16 cells incubated with Rho123 and **4** (40 μM) for 30 min. Control, untreated cells stained with Rho123. * *p* < 0.05; ** *p* < 0.01, n.s.—not significant.

**Figure 4 molecules-29-04939-f004:**
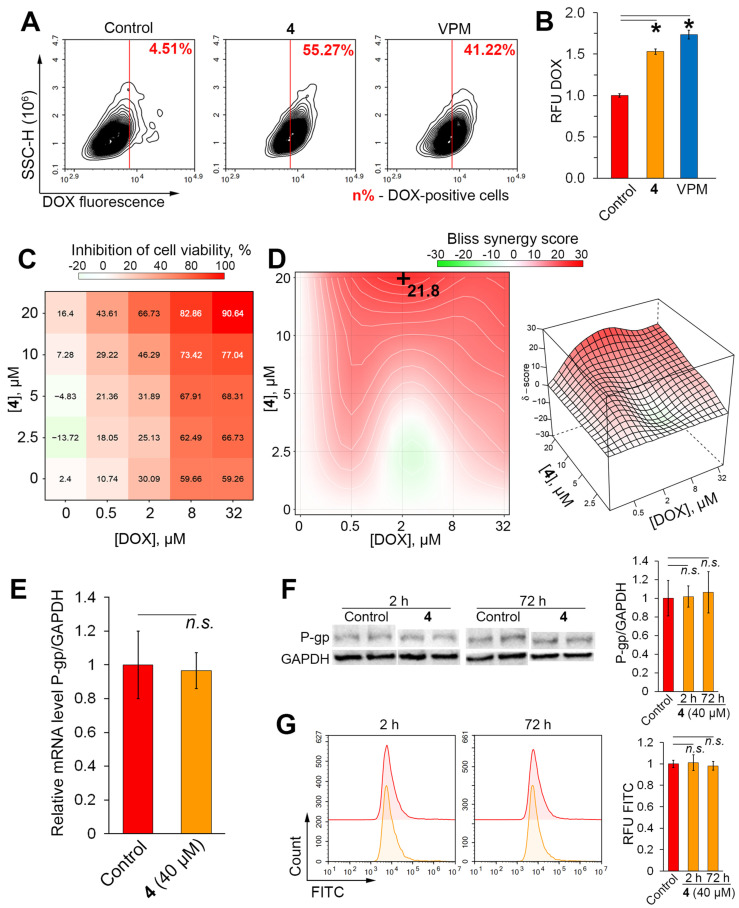
Effects of soloxolone amide **4** on doxorubicin (DOX) accumulation and MDR reversal in KB-8-5 cells. (**A**) Flow cytometry analysis of DOX fluorescence of KB-8-5 cells after incubation with DOX, VPM (50 μM), and **4** (40 μM) for 2 h. Control, untreated cells stained with DOX only. (**B**) Relative fluorescence level of KB-8-5 cells incubated with DOX, VPM (50 μM), and **4** (40 μM) for 2 h. Control, untreated cells stained with DOX only. (**C**) Combined effect of **4** and DOX on viability of KB-8-5 cells. Cytotoxicity rate was assessed by MTT assay after incubation with 4 (0–20 μM) and DOX (0–32 μM) for 72 h. (**D**) Two-dimensional and three-dimensional synergy plots of DOX and **4**-treated KB-8-5 cells. Bliss synergy score was calculated using SynergyFinder tool. (**E**) Relative mRNA level of ABCB1/GAPDH in KB-8-5 cells after incubation with **4** (40 μM) for 72 h evaluated by qRT-PCR. Control, untreated cells. (**F**) Western blot analysis of P-glycoprotein levels in KB-8-5 cells treated with **4** (40 μM) for 2 and 72 h. Control, untreated cells. (**G**) Flow cytometry assessment of P-gp expression in KB-8-5 cells treated with **4** (40 μM) for 2 and 72 h. * *p* < 0.05, n.s.—not significant.

**Figure 5 molecules-29-04939-f005:**
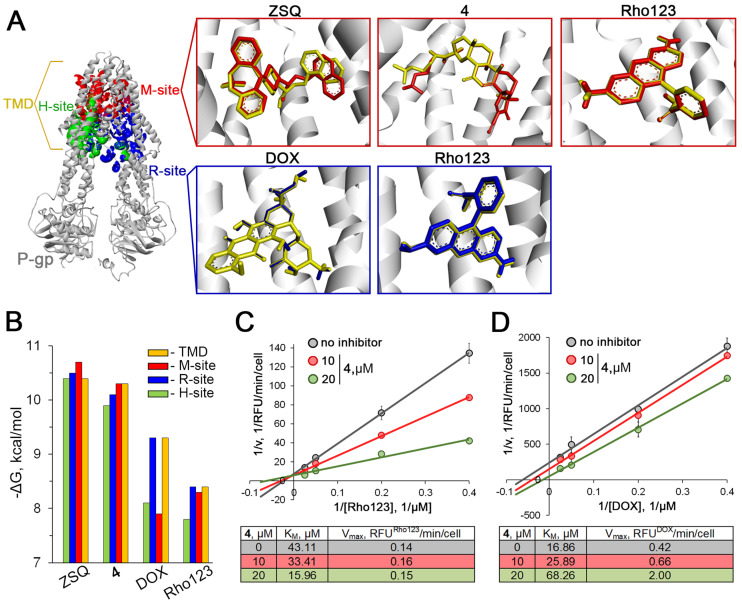
Investigation of mechanism of inhibition of P-gp transport activity by soloxolone amide **4** in KB-8-5 cells. (**A**) Top-ranked binding poses of ligands (ZSQ, **4**, Rho123, and DOX) within site-directed (R- and M-sites) and blind docking (to full transmembrane domain (TMD) of P-gp) to P-gp. Results of site-directed docking to R- and M-sites of P-gp correspond to blue- and red-colored molecules, respectively. Results of blind docking correspond to yellow-colored molecules. (**B**) Binding energies of ZSQ, **4**, Rho123, and DOX interacting with entire transmembrane domain and R-, H-, and M-sites of P-gp. (**C**) Lineweaver–Burk plot of Rho123 (2.5–40 μM) accumulation in KB-8-5 cells after 30 min incubation with **4** (10 or 20 μM) or without **4** (control) and calculated nonlinear kinetic parameters. V_accumulation_ = RFU^Rho123^/min/n_cells_. (**D**) Lineweaver–Burk plot of DOX (2.5–40 μM) accumulation in KB-8-5 cells after 150 min incubation with **4** (10 or 20 μM) or without **4** (control) and calculated nonlinear kinetic parameters. V_accumulation_ = RFU^DOX^/min/n_cells_.

**Figure 6 molecules-29-04939-f006:**
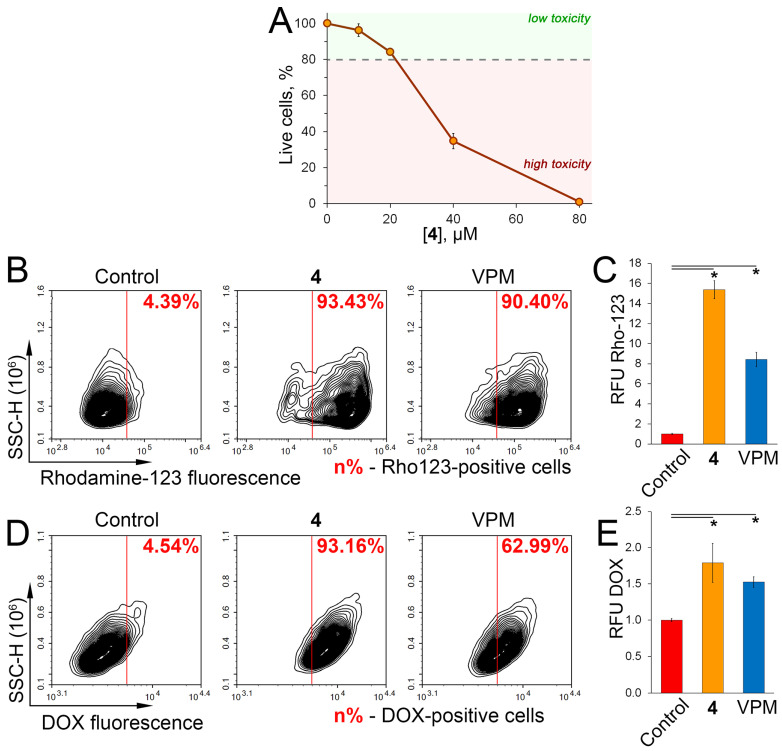
Validation of P-gp inhibitory activity of **4** in RLS40 cells. (**A**) Cytotoxicity of **4** (10–80 μM) in RLS40 cells evaluated by MTT assay after 72 h of incubation. (**B**) Flow cytometry analysis of Rho123 fluorescence of RLS40 cells after incubation with Rho123, VPM (50 μM), and **4** (20 μM) for 30 min. Control, untreated cells stained with Rho123 only. (**C**) Relative fluorescence level of RLS40 cells incubated with Rho123, VPM (50 μM), and **4** (20 μM) for 30 min (n = 3). Control, untreated cells stained with Rho123. (**D**) Flow cytometry analysis of DOX fluorescence of RLS40 cells after incubation with DOX, VPM (50 μM), and **4** (20 μM) for 2 h. Control, untreated cells stained with DOX only. (**E**) Relative fluorescence level of RLS40 cells incubated with DOX, VPM (50 μM), and **4** (20 μM) for 2 h (n = 3). Control, untreated cells stained with DOX. * *p* < 0.05.

**Table 1 molecules-29-04939-t001:** Sequences of primers for RT-PCR.

Gene	Type	Title 3
*ABCB1*	Forward	5′-AATGGCTACATGAGAGCGGAG-3′
	Reverse	5′-AATGTTCTGGCTTCCGTTGC-3′
*GAPDH*	Forward	5′-ACCCCCAATGTGTCCGTCGT-3′
	Reverse	5′-TACTCCTTGGAGGCCATGTA-3′

## Data Availability

Upon reasonable request, the corresponding author will provide the data generated and/or analyzed during this study.

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
