# Peer review of "Soloxolone N-3-(Dimethylamino)propylamide Restores Drug Sensitivity of Tumor Cells with Multidrug-Resistant Phenotype via Inhibition of P-Glycoprotein Efflux Function"

_molecules, 2024, doi:10.3390/molecules29204939_

Round 1
Reviewer 1 Report
Comments and Suggestions for Authors
The work by Moralev et al shows that compound 4, derived from soloxolone amides, inhibits drug efflux associated with P-gp (MDR1, ABCB1), can be delivered in nontoxic doses and cooperates with doxorubicin to induce cell death. The current work focuses on molecular modeling and in silico evaluation of the interaction of compound 4 with P-gp. The data presented was sound and properly interpreted. The methods were described in detail. While I do not suggest any additional assays at this time, it would be of interest if the authors would address a few points in their discussion/shortcomings.
1) Is there any interaciton of compount 4 with other members of the ABC family? Is there sufficient similarity/difference between the family members to predict is such interactions are/are not predicted?
2) Have the authors tested compound 4 in normal cells for toxicity and possible sensitization to drugs?
3) What is the impact of compound 4 on tumor cells that have not been trained to be drug resistant? In other words, cells that do not harbor MDR1 amplification.
4) Does compound 4 alter stem cell activity? A possibility since P-gp is associated with stemness.
5) Can the synthesis of compound 4 be scaled up? What are the challenges and expected cost? What is the time line necessary to produce enough compound 4 for in vivo application?
Author Response
Dear Reviewer #1,
We are very grateful for your detailed review of our manuscript, your careful attention to all described experiments, and your valuable comments, which have greatly improved our work. We have revised the manuscript according to your comments and would like to respond to your remarks.
1) Is there any interaciton of compount 4 with other members of the ABC family? Is there sufficient similarity/difference between the family members to predict is such interactions are/are not predicted?
Authors: Corrected. Dear Reviewer #1, Thank you for this important question. Indeed, other ABC family transporters are also involved in the development of multidrug resistance, and thus a study of the modulating effect of hit compound 4 on their pump activity would be of great interest. Considering the published data, to the best of our knowledge, each of the 49 known members of the human ABC family has its own substrate specificity, therefore, to evaluate compound 4-mediated direct blockade of non-P-gp efflux transporters, a number of conditions need to be fulfilled, including (a) selection of relevant cellular models with overexpression of the required proteins, (b) selection of fluorescent probes specific for the transporter of interest, and (c) additional molecular modeling, which is challenging for the present study. Given the importance of this issue, the need to evaluate the effect of 4 on the pump activity of other ABC family proteins has been added in section 2.6. Limitations of the study (see p. 11, lines 318-321).
2) Have the authors tested compound 4 in normal cells for toxicity and possible sensitization to drugs?
Authors: Corrected. In the present study, the key criterion for the selection of working concentrations of the tested triterpenoids was the absence of pronounced cytotoxic effects against model tumor cells (Fig. 3A, G, Fig. 6A). Evaluation of the safety of hit compound 4 for both non-transformed cells and non-tumor P-gp overexpressing tissues (such as kidney, liver, blood-brain barrier) in mice is the task of our further study in which we want to evaluate the probable anti-P-gp activity of 4 in vivo. Given the importance of the question you raise, the need for such studies has also been included in Section 2.6 Limitations of the Study (see pp. 10-11, lines 315-318).
3) What is the impact of compound 4 on tumor cells that have not been trained to be drug resistant? In other words, cells that do not harbor MDR1 amplification.
Authors: Corrected. Dear Reviewer #1, we are very grateful for this valuable comment. While investigating the anti-P-gp activity of 4, we completely forgot the control experiment with tumor cells that do not carry the MDR phenotype. To correct this, additional experiments were performed on Rho123 accumulation in compound 4-treated non-MDR tumor cells, namely human cervical carcinoma KB-3-1, human lung adenocarcinoma A549 and murine melanoma B16 cells. As shown in Figure 3H, I, compound 4 did not lead to a significant increase in the intracellular accumulation of Rho123 in these cells, confirming the P-gp targeting potential of 4. A description of the results obtained from these experiments and details on the cell lines used have been added to the manuscript (see p. 6, lines 163-166 and lines 189-192; p. 11, lines 334-336, 338). Thank you very much!
4) Does compound 4 alter stem cell activity? A possibility since P-gp is associated with stemness.
Authors: In our recently published report, we have indeed shown that soloxolone amides, in particular the para-methylanilide bearing compound 6, have a pronounced inhibitory effect on glioblastoma cell stemness, including clonogenicity and ALDH activity (Odarenko et al., 2024). Unfortunately, in the context of the present study, we did not evaluate the effect of 4 on stemness characteristics of KB-8-5 and RLS40 cells, being limited to relatively standard experiments in the context of anti-P-gp inhibitors. Given the link between stemness and P-gp activity noted by Reviewer #1, screening for the anti-stemness potential of soloxolone amides may be a relevant extension of this work. Thank you for this idea!
5) Can the synthesis of compound 4 be scaled up? What are the challenges and expected cost? What is the time line necessary to produce enough compound 4 for in vivo application?
Dear Reviewer #1, thank you very much for this important question, the answer to which is related to our future in vivo experiments. Since our article does not discuss the chemical part, we decided not to include in the manuscript any details regarding the availability of soloxolone N-3-(dimethylamino)propylamide, but in response to your question we would like to emphasize the commercial feasibility and scalability of the synthesis of 4. The starting compound for the synthesis of 4 is soloxolone methyl (Markov et al., 2022), which in turn is the product of a 10-step modification of 18βH-glycyrrhetinic acid (Logashenko et al., 2011).
Of course, the main problem in the synthesis of soloxolone derivatives is related to its multistep format, which requires time and highly qualified personnel (attentiveness, accuracy, experience with triterpenoids, etc.). However, considering the fact that 18βH-glycyrrhetinic acid (the starting material) is one of the cheapest compounds in its class (you can compare the prices of glycyrrhetinic, oleanolic and ursolic acids) and high availability of other reagents for synthesis (for example, available and cheap acetic and hydrochloric acids, hydrogen peroxide, hydroxylamine, etc.), the synthesis of soloxolone derivatives can be performed in a relatively short time (ethyl format).
We already had experience with the scale-up of soloxolone methyl in the amount of hundreds of grams from the ammonium salt of glycyrrhizic acid from licorice (purity ~ 70%). This procedure required time (about 5 months) and specific equipment to work with large volumes in the initial stages; the final stages were carried out in the laboratory. Thus, in terms of the reagents used, soloxolone methyl is not prohibitively expensive, and the main problems in scaling up are related to the high personnel requirements. Today, we can produce needed soloxolone amides for in vivo experiments, but in limited quantities.
We hope that corrected version of the manuscript will be acceptable for publication in the Molecules.
Sincerely,
On behalf of all authors,
Dr. Andrey Markov
References
Logashenko, E. B., Salomatina, O. V., Markov, A. V., Korchagina, D. V., Salakhutdinov, N. F., Tolstikov, G. A., et al. (2011). Synthesis and Pro-Apoptotic Activity of Novel Glycyrrhetinic Acid Derivatives. ChemBioChem 12, 784–794. doi:10.1002/cbic.201000618.
Markov, A. V, Ilyina, A. A., Salomatina, O. V, Sen’kova, A. V, Okhina, A. A., Rogachev, A. D., et al. (2022). Novel Soloxolone Amides as Potent Anti-Glioblastoma Candidates: Design, Synthesis, In Silico Analysis and Biological Activities In Vitro and In Vivo. Pharmaceuticals 15, 603. doi:10.3390/ph15050603.
Odarenko, K. V, Sen’kova, A. V, Salomatina, O. V, Markov, O. V, Salakhutdinov, N. F., Zenkova, M. A., et al. (2024). Soloxolone para-methylanilide effectively suppresses aggressive phenotype of glioblastoma cells including TGF-β1-induced glial-mesenchymal transition in vitro and inhibits growth of U87 glioblastoma xenografts in mice. Front. Pharmacol. 15, 1428924. doi:10.3389/fphar.2024.1428924.

Reviewer 2 Report
Comments and Suggestions for Authors
The manuscript entitled “Soloxolone N-3-(dimethylamino)propylamide Restores Drug Sensitivity of Tumor Cells with Multidrug Resistant Phenotype via Inhibition of P-glycoprotein Efflux Function” provide an interesting research and outcome to the scientific community. Authors discuss about the drug resistance, a hot issue now a days and the role of the P glycoprotein in the development of the resistance. The increase in the drug resistance and uncontrolled and inappropriate used of the drug decrease the sensitivity toward different drugs. The data provided very insightful, but some important point needs some clarification and scientific reasoning. Therefore, manuscript should be revised thoroughly before accepting for the publication. Therefore, it will gather the attention of wide spectrum of audience from the scientific community. Some points are indicated below for the consideration.
1. Different cell lines were used and use the different concentration of the compound, then how we simulate the findings for the comparison?
2. In general, the manuscript is well written but there are some minor grammatic mistakes, punctuations and typos in the manuscript that require a revision.
3. In the MTT assay, you incubate the cells for 30 min or 72 hours. Why this difference, it is highly significance difference?
4. Why you used the different assay for the cell viability in the different cell lines?
5. How you select the doses of doxorubicin to used in the experiment for the combine effects?
The authors should include the suggested modifications. Therefore, it is recommended that the manuscript should be revised before to accept the paper for publication in the journal. After the revision, the paper should be accepted for the publication in the journal.
Comments on the Quality of English Languagequality is good, just minor revision is required
Author Response
Dear Reviewer #2,
We are very grateful to you for your careful review of our manuscript and your extremely valuable comments, which have greatly improved our work. We have revised the manuscript according to your comments, and we would like to respond to them point by point.
- Different cell lines were used and use the different concentration of the compound, then how we simulate the findings for the comparison?
Authors: Corrected. Indeed, in two independent cell experiments we used different concentrations of the hit compound 4 (40 and 20 µM for KB-8-5 and RLS40 cells, respectively), which was determined by (a) different sensitivity of these cells to the tested triterpenoid and (b) the necessity to study the anti-P-gp activity of 4 at a non-toxic concentration to avoid false-positive results. The different sensitivity of KB-8-5 and RLS40 cells to 4 may be related to a different density of P-gp on the surface of these cells or to a possible more pronounced specificity of 4 for murine P-gp compared to the human transporter, which requires further detailed studies. In any case, the data obtained confirm that 4 is able to effectively block the function of the P-gp pump regardless of the organism (human or mouse). This text has been added to Section 2.5 (see p. 9-10, lines 290-296).
- In general, the manuscript is well written but there are some minor grammatic mistakes, punctuations and typos in the manuscript that require a revision.
Authors: Corrected. Dear Reviewer #2, we are very grateful for this comment. The text of the manuscript has been carefully reviewed by several authors and a number of unfortunate typos and errors have been corrected. Please see lines 26, 72, 73, 77, 83, 99, 99, 115, 116, 120-123, 140, 167, 167, 186, 203, 205, 204, 205, 242, 246, 257, 260, 308, 328, 348, 362, 364.
- In the MTT assay, you incubate the cells for 30 min or 72 hours. Why this difference, it is highly significance difference?
Authors: Corrected. Dear Reviewer #2, thank you for pointing out this error in the text of the manuscript, which could lead to misunderstanding on the part of the reader. The choice of such different time points was determined by the specifics of the experiments. The 30-min incubation was used to evaluate the non-toxic concentration of compounds (Fig. 3A) for further Rho123 assays, which according to generally accepted methods takes exactly 30 min (Phondeth et al., 2024; Rizvi et al., 2024), while the 72 h incubation was used to assess the combined cytotoxic activity of 4 and doxorubicin (DOX), as this time period is required to achieve significant DOX cytotoxicity, as we have previously shown (Moralev et al., 2023). This information has been added to the manuscript (see p. 5, lines 139-140, p. 6, line 204).
- Why you used the different assay for the cell viability in the different cell lines?
Authors: Corrected. Since RLS40 cells are characterized by low adhesion to culture plastic, we decided to replace the classical MTT assay with its analog (water-soluble WST-1 assay), which does not require removal of the medium for subsequent dissolution of the formazan crystals, where a significant loss of RLS40 cells is possible. This clarification has been added to the manuscript (see p. 11, lines 359-360).
- How you select the doses of doxorubicin to used in the experiment for the combine effects?
Authors: Corrected. As mentioned above (response to comment #3), the concentration range of DOX given in the manuscript was chosen for the combined experiment based on our previous findings (Moralev et al., 2023). In this range, the cytotoxicity of DOX varied from extremely low values to values below its IC50, which allowed us to clearly determine the presence of a synergistic association between 4 and DOX. A reference to our previous article in which a working range of DOX concentrations was determined has been added to the manuscript (see p. 6, line 204). Thank you for your attention to detail in the manuscript.
We hope that this version of the manuscript will be acceptable for publication.
Thank you very much!

Reviewer 3 Report
Comments and Suggestions for Authors
This manuscript presents a well-executed and timely study that explores the potential role of Soloxolane amide as a modulator of P-gp. Authors provide compelling evidence that compound 4 of Soloxolane amide showing a potential agent and may show effect to cure the MDR cancer, from this point of view this article present certain interest. This is a well-executed study, I have only one question that why authors show results of only MDR phenotype cells (both KB-8-5 and RLS40)? If they show according results on these cells having normal phenotype, will definitely enhances the magnitude of these findings. May be normal and MDR cells of both KB-8-5 and RLS40 these cell type shows the same kind of response. The authors should conform it.
Author Response
Dear Reviewer #3,
Thank you for taking the time to read and thoroughly analyze our article. We revised the manuscript according to your highly valuable comment, and, please, let us respond to it.
I have only one question that why authors show results of only MDR phenotype cells (both KB-8-5 and RLS40)? If they show according results on these cells having normal phenotype, will definitely enhances the magnitude of these findings. May be normal and MDR cells of both KB-8-5 and RLS40 these cell type shows the same kind of response. The authors should conform it.
Authors: Corrected. Indeed, while preparing this article, we completely forgot about the control experiment with non-MDR cells. To correct this oversight, the effect of 4 on Rho123 accumulation in tumor cells without MDR phenotype, including human cervical carcinoma KB-3-1 (non-MDR analog of KB-8-5 cells), human lung adenocarcinoma A549, and murine melanoma B16 cells, was evaluated. As shown in Figure 3H, I, 4 did not lead to an increase in the intracellular accumulation of Rho123, confirming the P-gp-targeting effect of the hit compound. Text describing the data obtained and information on the non-MDR cell lines used has been added to the manuscript (see p. 6, lines 163-166; p. 6, lines 189-192; p. 11, lines 334-336). We are very grateful to Reviewer #3 for this very valuable comment.
We hope that this version of the manuscript will be acceptable for publication in Molecules.
Thank you very much!

Reviewer 4 Report
Comments and Suggestions for Authors
Dear Author,
I give you the following comment, which is addressed in your manuscript to enhance the understanding and readability of the researcher.
Major Comments
1. Clarity of Objectives: Although the abstract gives a broad summary of the study, it does not clearly explain the main goals. It would be clearer if the primary study question or hypothesis were stated outright at the outset.
2. Methodological Detail: Although in vitro tests and molecular docking are mentioned in the abstract, there are no specifics about the employed methodology. Reproducibility and believability would be improved by providing a brief description of the experimental setup, including the parameters for the docking studies and the particular conditions for the assays.
3. Interpretation of the Results: Although the results are given, they are not given enough context. An example of this would be to contextualise the relevance of the binding energy (-10.3 kcal/mol). What does this indicate about the type and strength of the interaction with P-gp?
4. Comparison with Current medications: Although the reference inhibitor verapamil is mentioned, the abstract fails to provide a useful comparison of soloxolone N-3-(dimethylamino)propylamide's efficacy with that of other current medications, such as verapamil. A succinct analysis of its possible benefits or drawbacks would be helpful.
5. Implications for Future Research: A comment on how these findings may affect clinical applications or future research would be beneficial to include in the abstract. The study's relevance would be increased by highlighting prospective directions for future research or the development of new treatments.
Minor Comments
1. Technical Terminology: For a wider audience, especially those readers who might not be experts in the subject, the phrase "multidrug resistant phenotype" should be shortened or explained.
2. The chemical is designated as "soloxolone N-3-(dimethylamino)propylamide" and "compound 4." This demonstrates consistency in compound naming. Maintaining name consistency across the abstract would facilitate reader understanding.
3. Quantitative Data: Including particular quantitative data will strengthen the case for the findings. Examples of such data are the fold change in sensitivity to DOX or the percentage increase in medication absorption.
4. Statistical Analysis: The statistical techniques utilised to confirm the findings are not mentioned in the abstract. A succinct explanation of the statistical methodology would improve the findings' dependability.
5. Potential Limitations: A more fair assessment of the research findings would be obtained by addressing potential study limitations, such as the specificity of P-gp inhibition or the application of results to other cancer types.
Best Regards
Author Response
Dear Reviewer #4,
We are very grateful to you for your careful review of our manuscript and your valuable comments, which have greatly improved our work. We have revised the manuscript according to your comments, and we would like to respond to them point by point.
- Clarity of Objectives: Although the abstract gives a broad summary of the study, it does not clearly explain the main goals. It would be clearer if the primary study question or hypothesis were stated outright at the outset.
Authors: Corrected. To improve the structure of the abstract, the main objective of the study has been added (see lines 16-19).
- Methodological Detail: Although in vitro tests and molecular docking are mentioned in the abstract, there are no specifics about the employed methodology. Reproducibility and believability would be improved by providing a brief description of the experimental setup, including the parameters for the docking studies and the particular conditions for the assays.
Authors: Corrected. Please see p. 1, lines 19, 20 and 22.
- Interpretation of the Results: Although the results are given, they are not given enough context. An example of this would be to contextualise the relevance of the binding energy (-10.3 kcal/mol). What does this indicate about the type and strength of the interaction with P-gp?
Authors: Corrected. We are very grateful to Reviewer #4 for this comment, as the information on the binding energy threshold may be unknown to many readers. According to this comment, the sentence "Considering that binding energies lower than -7 kcal/mol indicate the formation of relatively stable protein-ligand complexes [33], the calculated ΔG values of the tested triterpenoids showed the relevance of their direct P-gp targeting potential" was added to the manuscript (see p. 3, lines 102-105).
- Comparison with Current medications: Although the reference inhibitor verapamil is mentioned, the abstract fails to provide a useful comparison of soloxolone N-3-(dimethylamino)propylamide's efficacy with that of other current medications, such as verapamil. A succinct analysis of its possible benefits or drawbacks would be helpful.
Authors: Corrected. Please see p. 1, lines 28-31.
- Implications for Future Research: A comment on how these findings may affect clinical applications or future research would be beneficial to include in the abstract. The study's relevance would be increased by highlighting prospective directions for future research or the development of new treatments.
Authors: Corrected. Please see p. 1, lines 32-35.
Minor comments
- Technical Terminology: For a wider audience, especially those readers who might not be experts in the subject, the phrase "multidrug resistant phenotype" should be shortened or explained.
Authors: Corrected. Please see pp. 1-2, lines 43-45.
- The chemical is designated as "soloxolone N-3-(dimethylamino)propylamide" and "compound 4." This demonstrates consistency in compound naming. Maintaining name consistency across the abstract would facilitate reader understanding.
Authors: Corrected. Indeed, different names for the same compound in the abstract can cause confusion for the reader. This problem has been fixed (see p. 1, lines 20-21).
- Quantitative Data: Including particular quantitative data will strengthen the case for the findings. Examples of such data are the fold change in sensitivity to DOX or the percentage increase in medication absorption
Authors: Corrected. Please see p. 1, lines 26, 27, and 30.
- Statistical Analysis: The statistical techniques utilised to confirm the findings are not mentioned in the abstract. A succinct explanation of the statistical methodology would improve the findings' dependability.
Authors: Corrected. Please see p. 1, lines 27-28.
- Potential Limitations: A more fair assessment of the research findings would be obtained by addressing potential study limitations, such as the specificity of P-gp inhibition or the application of results to other cancer types.
Authors: Corrected. The chapter on the limitations of the study has been significantly expanded. See pp. 10-11, lines 315-323.
We hope that this version of the manuscript will be acceptable for publication in Molecules.
Thank you very much!
